

# A new global anthropogenic SO₂ emission inventory for the last decade: A mosaic of satellite-derived and bottom-up emissions

Fei Liu[1,2], Sungyeon Choi[2,3], Can Li[2,4], Vitali E. Fioletov[5], Chris A. McLinden[5], Joanna Joiner[2], Nickolay A. Krotkov[2], Huisheng Bian[2,6], Greet Janssens-Maenhout[7], Anton S. Darmenov[2], Arlindo M. da Silva[2]

[1]Universities Space Research Association (USRA), GESTAR, Columbia, MD, USA
[2]NASA Goddard Space Flight Center, Greenbelt, MD, USA
[3]Science Systems and Applications Inc., Lanham, MD, USA
[4]Earth System Science Interdisciplinary Center, University of Maryland, College Park, MD, USA
[5]Air Quality Research Division, Environment and Climate Change Canada, Toronto, ON, Canada
[6]Goddard Earth Sciences and Technology Center, University of Maryland, Baltimore, MD, USA
[7]European Commission, Joint Research Centre, Institute for Environment and Sustainability, Via Fermi, Ispra (VA), Italy

*Correspondence to*: Fei Liu (fei.liu@nasa.gov)

**Abstract.** Sulfur dioxide (SO₂) measurements from the Ozone Monitoring Instrument (OMI) satellite sensor have been used to detect emissions from large point sources. Emissions from over 400 sources have been quantified individually based on OMI observations, accounting for about a half of total reported anthropogenic SO₂ emissions. Here we report a newly developed emission inventory, OMI-HTAP, by combining these OMI-based emission estimates and the conventional bottom-up inventory, HTAP, for smaller sources that OMI is not able to detect. OMI-HTAP includes emissions from OMI-detected sources that are not captured in previous leading bottom-up inventories, enabling more accurate emission estimates for regions with such missing sources. OMI-HTAP SO₂ emissions estimates for Persian Gulf, Mexico, and Russia are 59%, 65%, and 56% higher than HTAP estimates, respectively, in year 2010. We have evaluated the OMI-HTAP inventory by performing simulations with the Goddard Earth Observing System version 5 (GEOS-5) model. The GEOS-5 simulated SO₂ concentrations driven by both HTAP and OMI-HTAP were compared against in situ measurements. We focus the validation on year 2010 for which HTAP is most valid and a relatively large number of in situ measurements are available. Results show that the OMI-HTAP inventory improves the model agreement with observations, in particular over the US, with the normalized mean bias decreasing from 0.41 (HTAP) to -0.03 (OMI-HTAP) for year 2010. Additionally, our approach offers the possibility of rapid updates to emissions from large point sources that can be detected by satellites. Simulations with the OMI-HTAP inventory capture the worldwide major trends of large anthropogenic SO₂ emissions that are observed with OMI. For example, correlation coefficients of the observed and modelled surface SO₂ in 2014 increase from 0.16 (HTAP) to 0.59 (OMI-HTAP) and the normalized mean bias dropped from 0.29 (HTAP) to 0.05 (OMI-HTAP), when we updated 2010 HTAP emissions with 2014 OMI-HTAP emissions in the model. Our methodology applied to OMI-HTAP can also be used



to merge improved satellite-derived estimates with other multi-year bottom-up inventories, which may further improve the accuracy of the emission trends.

# 1 Introduction

Sulfur dioxide (SO$_2$) plays an important role in the Earth's ecosystems. As the principal precursor of sulfate aerosols, SO$_2$ has a significant effect on global and regional climate by changing radiative forcing (Seinfeld and Pandis, 2006) and degrading visibility (Cass et al., 1979). In addition, SO$_2$ emissions contribute to acid deposition that damages aquatic and terrestrial ecosystems. Anthropogenic SO$_2$ emissions, in particular those from the combustion of fossil fuels, are substantially greater than natural ones on a global basis (Smith et al., 2001) owing to the high concentrations of sulfur

contained in fossil fuels. In response to the rapid growth in fuel consumption driven by economic development in developing countries, particularly China, India, and international shipping, global SO$_2$ emissions increased from 2000 to 2005 (Smith et al., 2011). Meanwhile, stricter environmental legislation has promoted the introduction of new emission control with the fuel quality directive and desulfurization end-of-pipe abatement, in particular earlier in the US and Europe (Crippa et al., 2016) and more recently in China (C. Li et al., 2017). Additionally, shipping emissions over the Sulphur Emission Control Areas

(SECA) reduced since 2005 following the International Convention for the Prevention of Pollution from Ships (MARPOL) Protocol, which further strengthened measures in 2012 and 2013 (Alföldy et al., 2013). This has led to a decline in global SO$_2$ emissions since about 2006 (Klimont et al., 2013).

      SO$_2$ emissions usually are estimated using a bottom-up mass balance method. Bottom-up emissions are equal to the amount of sulfur in the fuel (or ore) minus that removed or retained in bottom ash or in products (Smith et al., 2011). The

magnitude of emissions is subject to uncertainties, particularly when information on sulfur contents of fuels/ores or sulfur removals is not available. The spatial distribution of emissions is more uncertain, as emissions within a region are allocated by spatial proxies in most cases rather than actual locations of emission sources owing to a dearth of data. In addition, SO$_2$ emission inventories developing for a specific year may become outdated if applied to other years when technologies and fuel use change rapidly.

25        SO$_2$ observations from space-based platforms provide valuable global information on the spatio-temporal patterns of SO$_2$ emissions (Krotkov et al., 2016) that may complement existing bottom-up emission inventories and help to indicate hot spots. Satellite-measured SO$_2$ has been used to monitor and characterize regional emission trends (van der A et al., 2017), volcanic emissions (Theys et al., 2013; Carn et al., 2016), and anthropogenic emissions from large point sources like smelters (Carn et al., 2007), power plants (Li et al., 2010), and oil sands (McLinden et al., 2012). Additionally, satellite

retrievals of SO$_2$ vertical column densities have been used to quantify the strength of SO$_2$ emissions (Fioletov et al., 2015; 2017).



Chemical transport models (CTMs) have been employed to exploit SO$_2$ observations as a constraint towards improving SO$_2$ inventories using inverse modeling techniques (Lee et al., 2011; Wang et al., 2016). However, the derived emissions are usually determined at the coarse spatial resolution of CTMs (e.g., 2 ° latitude by 2.5 ° longitude in Lee et al., 2011) and are subject to large uncertainties at finer spatial scales. Alternative CTM−independent approaches have been

proposed to resolve SO$_2$ signals around individual large sources with simple model functions such as Gaussian (Fioletov et al., 2011). More recently, SO$_2$ emission rates and lifetimes were fitted simultaneously from the satellite-observed downwind plume evolution and meteorological wind fields for volcanoes (Beirle et al., 2014) and anthropogenic sources (e.g., Fioletov et al., 2015; 2017).

The satellite-based approaches used to estimate emissions are generally limited to larger sources, typically > 30

Gg/yr (Fioletov et al., 2016), for the highest spatial resolution observations currently available from the Ozone Monitoring Instrument (OMI). Here, we develop a methodology to provide a comprehensive emission inventory that combines large SO$_2$ source information from satellite-derived emissions with the conventional bottom-up emission estimates for smaller sources. An overview of the satellite-derived and the bottom-up inventories used in this study is provided in Sections 2.1 and 2.2, respectively. The methodology and features developed for our merged inventory are detailed in Sect. 2.3. Section 3 describes

the model and in situ measurements used for evaluating our merged inventory, respectively. Section 4 details the validation results. The validation focuses on year 2010 for which the bottom-up inventory used by this study is most valid and a large number of in situ measurements are available. The validation for other years is performed to evaluate the emission trend of large sources that can be detected by OMI. Section 5 compares our inventory with other existing bottom-up inventories. Section 6 presents a summary of the performance of the new inventory and the future work plans for maintaining and

improving the inventory.

## 2 Emissions

### 2.1 Satellite-derived emission inventory

The global OMI measurements allow for quantification of SO$_2$ emissions from anthropogenic sources. OMI is a UV-VIS nadir-viewing satellite spectrometer (Levelt et al., 2006, 2017) on board the NASA Aura spacecraft launched in 2004. We

use the OMI-based emission catalogue of nearly 500 sources from Fioletov et al. (2016) to develop a new global SO$_2$ emission database in this study. The OMI-based emission catalogue is based on version 1.3 level 2 (orbital level) OMI planetary boundary layer (PBL) SO$_2$ products retrieved with the principle component algorithm (PCA) algorithm (Li et al., 2013) and the updated Air Mass Factors (AMFs) for each site (McLinden et al., 2014). The OMI SO$_2$ observations are rotated according to wind directions such that all observations were aligned in one direction (from upwind to downwind;

Valin et al., 2011; Fioletov et al., 2015). The location of the source is derived by comparing the difference between the average downwind and average upwind SO$_2$ column (McLinden et al., 2016). The rotated observations are assumed to be a single point source convolved with a Gaussian function (Beirle et al., 2014) and fitted by a three-dimensional



parameterization function of horizontal coordinates and wind speeds (Fioletov et al., 2015) in order to estimate emissions. Only observations contained within a rectangular area (hereafter called the fitting domain) are used for the fit. The fitting domain spreads ±Ł km across the wind direction, L km in the upwind direction and 3 Ł km in the downwind direction. The value of L is chosen to be 30 km for small sources (under 100 Gg/yr), 50 km for medium sources (between 100 and 1000

Gg/yr), and 90 km for large sources (more than 1000 Gg/yr). Note that we prescribe values of the lifetime and the parameter describing the spread of the emission plume to obtain more robust fitting results. Additional information on the algorithm and uncertainties in the emissions are available from Fioletov et al. (2016). In this way, the emissions and site coordinates are directly derived from OMI observations. The source types are further authenticated through a combination of satellite imagery and external databases based on site coordinates. The annual $SO_2$ emission, site coordinate, source type (power

plant, smelter or source related to the oil and gas industry) for each anthropogenic source in the catalogue for the period from 2005 to 2014 are used here.

## 2.2 Bottom-up emission inventory HTAP

We use the up-to-date global anthropogenic emission inventory developed by the Task Force Hemispheric Transport Air Pollution (HTAP v2.2, available at http://edgar.jrc.ec.europa. eu/htap_v2) for sources that satellites are unable to detect. The

HTAP v2.2 emission database is a state-of-art inventory compiling the latest available official and regional emission data and has been widely used in global and regional modelling experiments (e.g., Bian et al., 2017; Paulot et al., 2016; Ojha et al., 2016). It provides annual and monthly gridded air pollutant emissions with global coverage at a spatial resolution of 0.1 ° × 0.1 °for the years 2008 and 2010 (Janssens-Maenhout et al., 2015).

The gridded HTAP v2.2 $SO_2$ emission maps are provided for six categories (energy, industry, residential, ground

transport, aviation, and shipping). Some of the emissions from the energy and industry sector are identified as point sources and allocated to their exact locations; others are treated as areal sources and distributed to grid cells based on spatial proxies due to the lack of information on locations. Emissions from large-scale biomass burning (including Savannah fires, field burning and forest fires) are excluded from the inventory, of which the share to the total $SO_2$ emissions is small and varies between 2.0% and 3.6% (for the period of 2005–2010) (EDGAR v4.2 and fast track updates of EC-JRC/PBL, 2011).

HTAP v2.2 is a mosaic emission database that merges emission grid maps from the US Environmental Protection Agency (EPA) and Environment and Climate Change Canada for North America (Pouliot et al., 2015), Monitoring Atmospheric Composition and Climate - Interim Implementation (MACC-II) for Europe (Kuenen et al., 2014), the 2012 version of MIX for Asia (M. Li et al., 2017), and the Emission Database for Global Atmospheric Research version 4.3 for the rest of the world (EDGAR v4.3; Crippa et al., 2016). Although the data provided in each inventory aims to actually

represent 2008/2010 at the spatial resolution of 0.1 °×0.1 °, the dataset was not consistently compiled with activity statistics of 2008/2010 (as is the case in EDGAR v4.3).

The National Emissions Inventory (NEI) of the US EPA is compiled bottom-up every three years and updated for the years in between with total consumption-based trends.  The 2010 data for the US are based on the 2008 NEI with year-



specific updates made for power plants equipped with continuous emissions monitoring system and on-road mobile sources (Pouliot et al., 2014); for other sources a trend has been applied based on the trend in the sector-specific country totals. The 2010 data for Canada are based on the 2008 National Emission Inventory of Environment and Climate Change Canada with updated emissions for point sources (Pouliot et al., 2014). The 2008 HTAP data for Europe are assumed to be the same as

the 2009 MACC-II data, and the 2010 HTAP data for Europe are derived by extrapolating the 2009 MACC-II data based on the trend in the MACC-II inventory between 2006 and 2009, as the MACC-II inventory is only available for 2006 and 2009 when developing HTAP (Janssens-Maenhout et al., 2015).

In addition, re-sampling is applied to obtain gridded maps with a uniform spatial resolution of $0.1\,° \times 0.1\,°$ based on the MACC-II inventory at $1/8\,° \times 1/16\,°$ resolution and the MIX inventory at $0.25\,° \times 0.25\,°$ resolution. As pointed out by the

HTAP report (Janssens-Maenhout et al., 2015), such inconsistency between the different inventories may yield uncertainties in strengths and locations of emissions. For example, emissions from large point sources with changing emission patterns cannot be accurately derived from a linear extrapolation in time, because such extrapolation is not able to reflect sudden changes, such as shutting down of certain sources.

## 2.3 OMI-HTAP harmonized emission inventory

The OMI-based and the HTAP emission inventories are merged to construct a harmonized inventory that we refer to as OMI-HTAP. OMI-HTAP is particularly developed for the years 2008 and 2010 when HTAP is available. For other years, emissions from large sources that can be dectected by satellites are updated in OMI-HTAP. For other sources including those from the aviation and shipping sectors, the 2008 HTAP v2.2 inventory is used for construction of the OMI-HTAP inventory for years prior to 2008 as well as 2009; similarly, the 2010 HTAP v2.2 is used for years after year 2010. The emissions from

these sources can be further updated using latest bottom-up inventories with multi-year estimates. Consistent with the HTAP inventory, the OMI-HTAP inventory provides monthly gridded $SO_2$ emissions with global coverage at a spatial resolution of $0.1\,° \times 0.1\,°$ for different sectors.

Figure 1 shows the schematic methodology of the OMI-HTAP emission inventory development. For each grid cell in the HTAP inventory, its emissions are replaced by OMI-based estimates if it is located inside the fitting domain of any

sources in the satellite-derived inventory; otherwise, its emissions remain to be combined with the OMI-based emissions. The OMI-based emissions for individual years are allocated to corresponding grid cells according to their coordinates. The emissions from power plants and other industrial facilities are categorized as emissions from the energy and industry sector in the OMI-HTAP inventory, respectively.

In order to estimate monthly emissions from OMI, its annual emissions are scaled by the HTAP monthly profiles

averaged over the fitting domain for the corresponding sector. That is, the OMI-based emissions are regarded as a single source within a particular fitting domain; areas not included within any fitting domain use HTAP emission grid maps.

Figure 2 displays the 2010 OMI-HTAP $SO_2$ inventory (top) and compares it with the 2010 HTAP inventory (bottom). The two inventories are consistent in total amount with a slightly higher (1%) estimate from the OMI-HTAP.



However, they differ in the spatial distribution of emissions. Reasonable agreement is found in total emissions over China and most Western and Central European countries (differing by 2–8%), while the discrepancies in locations of emissions are shown. Consistent with the findings in McLinden et al. (2016), higher OMI-HTAP estimates cluster over the Persian Gulf, Mexico, and Russia, with the OMI-HTAP $SO_2$ emissions estimates 59%, 65%, and 56% higher, respectively. On the contrary, lower OMI-HTAP estimates are concentrated over US and India, with OMI-HTAP estimates 31% lower.

Uncertainties in the OMI-based estimates may contribute to the differences. These uncertainties primarily arise from the air mass factor calculation, noise in OMI measurements, and the emission fitting procedure (Fioletov et al., 2016). On the other hand, uncertainties inherent in the total magnitude and the spatial distribution of bottom-up emissions may also contribute to the differences such as when bottom-up emissions are not routinely updated. In fact, emissions from some emitting sectors in bottom-up inventories are not tracked with individual point sources but spread out over larger areas instead. The country-specific emissions in HTAP are allocated where possible to the locations of point sources (e.g. public electricity plants), but a large fraction (e.g. some smelters of which the location are not available) remains distributed over the countries with spatial proxies (e.g., urban population) of which the representativeness is only qualitatively known.

Bottom-up US $SO_2$ estimates are considered to be accurate, as over half of the emissions are directly measured by continuous emission monitoring systems. However, the emissions from the source types without continuous monitoring devices, including some power plants (ranging from 10–20% for the period of 2005–2014; US EPA, 2014) as well as other industrial and residential sources were not tracked as point sources in HTAP, but distributed over a larger area making use of spatial proxies. Moreover, updates on the fuel quality and technologies in these sources since 2008 were not accounted for. The discrepancy over the US is most likely related to such sources.

HTAP estimates 9% and 12% declines of $SO_2$ emissions for energy and industry sectors in the US, respectively, from 2008 to 2010; this is less than the reported 27% and 20% decline by EPA (EPA Air Pollutant Emissions Trends Data; available at https://www.epa.gov/air-emissions-inventories/air-pollutant-emissions-trends-data); HTAP estimates are higher than the OMI-HTAP estimate for year 2010. For year 2008 with better information on the fuel quality and technologies in HTAP, the discrepancy between the two inventories over the US is much less (17%). This is further supported by the excellent agreement for the largest individual US sources, for which emissions are based on direct stack measurements using continuous emission monitoring systems (Figure 3, Fioletov et al., 2015; Figure 1, Fioletov et al., 2017).

In other regions, uncertainties in bottom-up inventories could be larger owing to the lack of local emission measurements including continuous emission monitoring. For instance, local emission measurements in India are sparse and discrepancies between estimates from different bottom-up inventories can be as large as 50% (M. Li et al., 2017). The sulfur content of Indian fossil fuels adopted by HTAP was based on assumptions in the MIX inventory. This inventory includes detailed information on China; however, there is much less information available for India owing to limited reporting in the literature (e.g., Reddy and Venkataraman, 2002). In addition, the fuel use is usually based on officially reported statistics, which may not be accurately reported. Some fuel consumption in South Asia is not included in official statistics, such as the burning of kerosene for wick lamps or fuel oil for diesel generators (Lam et al., 2012), which may be even more uncertain.



Long-standing experience in the development of emission inventories suggests that bottom-up inventories may miss some significant sources. The higher values over the Middle East, Mexico, and Russia in OMI-HTAP are due to the inclusion of emissions from the OMI-identified sources missing from HTAP (McLinden et al., 2016). This helps to make OMI-HTAP a more complete inventory for these regions.

The locations of emissions in HTAP sometimes deviate from those in OMI-HTAP. This is probably caused by different geographical allocation methods in two inventories, in particular the use of spatial proxies instead of real point source locations. In the OMI-based estimates, the location of each individual source is obtained from the OMI observations and then manually verified with satellite images in Google Earth; this can lead to high accuracy. In the HTAP inventory, spatial proxies like total, rural, and urban population densities, road network and combinations were adopted to downscale

emissions that lack geographical information; this may produce uncertainties when emission locations are decoupled from spatial proxies (Liu et al., 2016, 2017). Section 6 provides further discussion regarding the spatial mismatch of emission sources in HTAP and OMI-HTAP.

## 3 Model and in situ measurements

### 3.1 GEOS-5 Model

We use the NASA Global Modeling and Assimilation Office (GMAO) Goddard Earth Observing System version 5 data assimilation system (GEOS-5 DAS) (Rienecker et al., 2008) to simulate global surface $SO_2$ in this study. The aerosol module in GEOS-5 is based on the Goddard Chemistry Aerosol Radiation and Transport (GOCART) model (Chin et al., 2002). The model simulation is driven by GMAO atmospheric analyses from the Modern-Era Retrospective Analysis for Research and Applications, version 2 (MERRA-2; Gelaro et al., 2017) in what is referred to as a replay mode where the aerosol fields do

not feed back to the system. In other words, we run the GEOS-5 aerosol module in forecast-mode with initial conditions from a previous run of the system, and the resulting aerosol fields do not impact the radiation within the model as they do in a full model run. The replay mode is run at a resolution of $0.5\,°\times0.5\,°$ and 72 vertical layers between the surface and about 80 km.

We ran the system using either the HTAP or OMI-HTAP inventory within the aerosol module. We allow a one-

month spin up of aerosol fields for each experiment. For both the HTAP and OMI-HTAP emissions, we allocate the non-energy emissions (from industrial, residential, and transportation sectors) to the lowest GEOS-5 layer and the energy emissions from power plants to levels between 100 and 500 m above the surface (Buchard et al., 2014). All the simulations include aircraft and ship emissions from the HTAP v2.2 inventory, biomass burning emissions from the Quick Fire Emission Dataset (QFED) inventory (van der Werf et al., 2010), production from dimethyl sulfide (DMS) oxidation (Kettle et al.,

1999). Volcanic $SO_2$ emissions are derived from Total Ozone Mapping Spectrometer (TOMS), OMI, and Ozone Mapping and Profiler Suite (OMPS) $SO_2$ retrievals (Carn et al., 2015) and the Aerocom inventories (Diehl et al., 2012).



While the main focus here is on year 2010, we also conducted GEOS-5 simulations for 2006 and 2014 in order to evaluate the trends detected by the satellite data. $SO_2$ concentrations are simulated based on the 2008 HTAP and the 2006 OMI-HTAP inventories for year 2006 the 2010 HTAP and the 2010 OMI-HTAP inventories for year 2010, and the 2010 HTAP and the 2014 OMI-HTAP inventories for year 2014.

5        Figure 3 illustrates the annual mean surface $SO_2$ simulation using both inventories for year 2010. Not surprisingly, the differences (Fig. 3b) show spatial patterns similar to the emission changes (Fig. 2b). The concentrations in the lowest model layer (from ground up to around 50 m) are evaluated using surface $SO_2$ observations in the following analysis.

## 3.2 $SO_2$ measurements used for evaluation

We evaluate the modelling surface concentrations of $SO_2$ over the US, Europe and East Asia for the years 2006, 2010 and 2014 using in-situ measurements from air quality networks. We use stations from the US EPA Air Quality System (AQS; available at https://www.epa.gov/aqs) for the US, the European air quality database (AirBase; available at https://www.eea.europa.eu) for Europe, and the Acid Deposition Monitoring Network in East Asia (EANET, available at http://www.eanet.asia) for East and Southeast Asia. For our analysis, we only include stations that had quality-controlled data for at least 75% days for an individual year. We further exclude stations located in mountainous regions with an elevation of over 1000 m, as we expect model limitations in describing pollutant concentrations over complex terrain (Liu et al., 2018). Additionally, we exclude stations located in regions with volcanoes as the dominant $SO_2$ source, e.g., Hawaii; the aim of this evaluation is to assess the performance of HTAP and OMI-HTAP, and volcanic emissions have not been considered in either inventory. This leaves 248, 818, and 32 stations across US, Europe, East Asia, respectively.

Sites in US-AQS and EU-AirBase are typically closer to urban areas. These sites may not be representative of the model grid-cell mean when impacted by local pollution. To increase representativeness of grid box values, the available in-situ measurements are averaged over the model's $0.5°\times0.5°$ grid cells before comparison with the model output.

## 4 Evaluation of the OMI-HTAP inventory

### 4.1 Model comparison to surface measurements in 2010

Figure 4 shows scatter plots of the modelled $SO_2$ driven by HTAP (left) and OMI-HTAP (right) versus in-situ measurements for the US (top), Europe (middle), and East and Southeast Asia (bottom) in 2010. The plots show considerable scatter between modelled and observed annual means with correlation coefficients of 0.53 and 0.50 over the US, 0.40 and 0.48 over Europe, and 0.60 and 0.53 over East and Southeast Asia for simulations with HTAP and OMI-HTAP respectively. Here, we focus on the differences between modelled $SO_2$ using the OMI-HTAP and HTAP inventories. However, we note that the scatter between modelled and observed values may be attributed to the representativeness error related to the incompatibility between in-situ measurements and grid-cell averaged values simulated by the model. In addition, the slightly longer $SO_2$ lifetime simulated by the model as compared with in situ measurements and uncertainties in emissions may further



contribute to the discrepancy (Buchard et al., 2014). Additional details on evaluation of GEOS-5 $SO_2$ simulations can be found in Buchard et al. (2014).

The implementation of OMI-HTAP improves the GEOS-5 performance with respect to observed surface $SO_2$ concentrations. We calculate normalized mean bias (NMB) to quantify the differences between modelled and observed $SO_2$

concentrations; NMB is defined as $\frac{\sum_1^n(M-N)}{\sum_1^n N}$, where $M$ and $N$ represent modelled and observed quantities, respectively.

The reduction in NMB for 2010 is highlighted for the US in Figure 4 (top) with values of 0.41 using HTAP and -0.03 using OMI-HTAP. The reduction is particularly significant for grid cells with emission changes when comparing the two inventories, with NMB values of 0.70 and 0.06 for simulations with HTAP and OMI-HTAP, respectively. Improvements in Europe and Asia are much more subtle; most observations are made in grid cells with no differences

between OMI and OMI-HTAP. For example, no significant changes are detected for Asia as most EANET sites are located far away from areas with modified emissions in OMI-HTAP.

Figure 5 further illustrates the spatial distribution of the 2010 differences by comparing the annual averaged $SO_2$ concentrations from the AQS measurements (a) and the GEOS-5 simulations together with HTAP (b) and OMI-HTAP (c). It identifies the considerable changes over the Eastern US that contributes to the US bias reduction using OMI-HTAP.

Simulated $SO_2$ with HTAP is overestimated for most stations without discernible seasonal variations (not shown), while the widespread overestimation is not observed in simulations with OMI-HTAP. The bias reduction of simulations over the US is attributed to the timely update of $SO_2$ emissions in OMI-HTAP. The magnitude of $SO_2$ emissions decreases by 25% in OMI-HTAP during 2008 to 2010, much higher than the decline of 9% in HTAP (see details in Sect. 2.3).

Improved agreement between observations and simulations is also shown for Europe. In particular, for grid cells

with emission changes, the correlation coefficient increases from 0.29 (HTAP) to 0.44 (OMI-HTAP). A plausible explanation for the improvement is the more reasonable spatial distribution of all large emission point sources in OMI-HTAP as detailed in Sect. 2.3.

## 4.2 Validation of emission trends in satellite data

In this section, we highlight the improvements obtained with OMI-HTAP for tracking emission changes driven by trends in

the OMI data. Global anthropogenic $SO_2$ emissions substantially decline in OMI-HTAP. The US, Europe, and China are the primary contributors to the emissions reductions, showing declines of 47%, 27%, and 23% in the OMI-HTAP $SO_2$ emissions during 2005–2014 respectively. These declines are attributed in part to the installation of flue-gas scrubbers for coal-fired power plants. In addition, emissions from the world's largest smelters decreased due to phase out of operations in some plants (e.g., Ilo, Peru; Flin Flon, Canada) or installation of scrubbers (e.g., La Oroya, Peru) (see more details in Sect.5.2 of

Fioletov et al., 2016). In contrast, India experienced a rapid rise in emissions with a growth of 39% in OMI-HTAP emissions during 2005–2014, potentially surpassing China as the world's largest emitter of anthropogenic $SO_2$ (C. Li et al., 2017).





The capability of OMI-HTAP (in particular OMI) to capture the emission trends is examined in Figure 6. We compare the GEOS-5 simulations using both HTAP (grey dots) and OMI-HTAP (blue dots) with in-situ surface measurements for years 2006 (Fig. 6a) and 2014 (Fig. 6b). The agreement between the observed and modelled $SO_2$ is better with simulations using OMI-HTAP, with higher correlations and lower biases. This is particularly true for year 2014 with a

large gap (i.e., 4 years) in the time for which emissions are developed between HTAP and OMI-HTAP. Correlation coefficients in 2014 increase from 0.16 (HTAP) to 0.59 (OMI-HTAP) and the normalized mean bias dropped from 0.29 (HTAP) to 0.05 (OMI-HTAP). The improvements arise from the updated emissions in OMI-HTAP, in particular the declines in emissions over the US and China from 2010 to 2014. The 2014 OMI-HTAP $SO_2$ emissions are 41% and 14% lower than 2010 HTAP estimates for the US and China, respectively. The better consistency with measurements for both years indicates

that OMI (and thus the OMI-HTAP inventory) captures changes in emissions during the 8-year span.

## 5 Intercomparison of bottom-up inventories

In this section, we compare OMI-HTAP with bottom-up emission inventories that are widely used within the climate and air-quality modelling community. The discussion is focused on inventories that are incorporated into HTAP (hereafter called incorporated inventories), including the global EDGAR v4.3 inventory (Crippa et al., 2016), the European MACC-II

inventory (Kuenen et al., 2014), and the Asian MIX inventory (M. Li et al., 2017). Two additional regional inventories, the European Monitoring and Evaluation Programme (EMEP, Mareckova et al., 2013) at $0.5° × 0.5°$ resolution and Regional Emission inventory in Asia version 2 (REAS 2, Kurokawa et al., 2013) at $0.25° × 0.25°$ resolution are taken into account; these are closely related to MACC-II and MIX, respectively. We use the year 2010 to conduct the comparison because this is the most recent year when emissions are available in all inventories with the exception of REAS 2. The year 2008 is chosen

for REAS 2, as emissions after 2008 are not available. The comparison is performed, focusing on OMI-detected large point sources, to highlight the new features of OMI-HTAP and to identify the potential sources of uncertainties in bottom-up inventories.

We first focus on emission locations. For each OMI-detected source, if the bottom-up estimate is less than 20% of the OMI-based estimate (out of the uncertainty range of satellite-derived emission estimates) in the fitting domain (see the

definition in Sect. 2.1), the source is considered to be missing from the bottom-up inventory. Otherwise, the location of the grid cell with the maximum emission within the fitting domain is identified to compare with that in the OMI-based emission catalogue (Fioletov et al., 2016) used by OMI-HTAP. A source found within the fitting domain is classified as matched when the locations in the OMI-based emission catalogue and the bottom-up inventory are the same; otherwise, the source is classified as relocated and the distance between the OMI-detected and the bottom-up inventory source is calculated. The

comparison is performed for four regions separately, i.e., North America, Europe, Asia, and the rest of the world (other). Note that emissions from countries that are only partly covered by the either the European or Asian inventories (e.g., Russia, Turkmenistan, Uzbekistan and Kazakhstan) are categorized as other in this study to stay consistent with HTAP.



Figure 7 summarizes the differences of emission locations between the OMI-based emission catalogue (and thus OMI-HTAP) and bottom-up inventories. HTAP shows the best agreement with OMI in North American, the region where it is expected to have good knowledge of large $SO_2$ emission sources in bottom-up. The average distance between sources in HTAP and the OMI-based emission catalogue is merely 4 km for North America. This is significantly less than the mean

distances differences of 20 km, 22 km and 15 km for Europe, Asia, and other regions, respectively.

It is interesting to note that sources are not always consistently located in HTAP and its incorporated inventories. The average mismatch of locations between the OMI-based emission catalogue and HTAP is significantly larger than that between the OMI-based emission catalogue and the incorporated inventories for both Europe (20 km for HTAP vs 12 km for MACC-II) and Asia (22 km for HTAP vs 17 km for MIX). The enhanced distances for HTAP are associated with a loss of

spatial accuracy by the upscaling of incorporated inventories to a coarser grid (e.g., MACC-II for Europe has a higher resolution than HTAP) and by the re-sampling of grids that are not a multiple of 0.1°. Re-sampling is applied to merge grid maps at different spatial resolution (i.e., $1/8° \times 1/16°$ for MACC-II and $0.25° \times 0.25°$ for MIX) to the common resolution of $0.1° \times 0.1°$ for HTAP (Janssens-Maenhout et al., 2015). This potentially misallocates emissions and thus increases the number of relocated sources (grey in Fig. 7).

Additionally, the incorporated inventories show better consistency in terms of location than other inventories developed for the same regions (i.e., EMEP for Europe and REAS for Asia) as compared with the OMI-based emission catalogue. For MACC-II, the improved consistency arises from its fine spatial resolution of $1/8° \times 1/16°$, higher than that of $0.5° \times 0.5°$ for EMEP. For MIX, the better consistency is attributed to the improved spatial patterns associated with the incorporation of local high-resolution emission datasets, such as the China coal-fired Power plant Emissions Database

(CPED, Liu et al., 2015) and an Indian emission inventory for power plants developed by Argonne National Laboratory (Lu et al., 2011).

We further examine individual sources with annual bottom-up $SO_2$ emissions exceeding 70 Gg/yr that are expected to produce a statistically significant signal in OMI data (Fioletov et al., 2011) but are not found in the OMI-based emission catalogue of nearly 500 sources (Fioletov et al., 2016). These large sources that are indicated by different bottom-up

inventories mentioned previously in this section are shown in Fig. 8b–e as solid and open circles for power plants and other types of sources, respectively. There are 74 such sources in total with 15 from HTAP, 31 from EDGAR, 3 from MACC-II, 14 from MIX, and 11 from REAS.

Bottom-up sources are likely not be seen by OMI if they are located in regions with large systematic bias and retrieval noise for OMI PBL $SO_2$ data. These conditions occur, for instance, at high latitudes and over the South Atlantic and

South America (from southern Peru southward) that are affected by the South Atlantic Anomaly that increases detector noise in OMI observations (Fig. 8c). Additionally, bottom-up sources located in close proximity to other significant sources like volcanoes (Indonesia in Fig. 8e) could be absent from the OMI-based emission catalogue, as OMI may have difficulty in separating emission signals from individual sources.



In general, information on emissions from large sources individually my not be consistent among bottom-up inventories; sources identified as significant in one inventory may be missing from another, depending on the quality of the point source database used as input. Bottom-up emissions from large point sources are derived from distributing country total emissions for the corresponding sector to individual facilities, when emissions at the facility level are not available.

Emissions from large sources are potentially represented with too strong intensity concentrated over a limited number of specific locations in the country. In this way, less point sources identified by bottom-up inventories in total lead to more sources with strong emission intensity, which may explain why more sources (31) in EDGAR are missing from the satellite-derived emission catalogue compared with those (15) in HTAP.

Figure 8f compares emissions from global/regional inventories considered in this section to those from unit-based

inventories for the power plants shown in Fig. 8b–e (solid circles). The considered unit-based power plant databases include Emissions & Generation Resource Integrated Database (eGRID) for the US (US EPA, 2014), CPED (Liu et al., 2015) for China, and the European Pollutant Release and Transfer Register (E-PRTR; available from https://www.eea.europa.eu/data-and-maps/data/lcp-4) for Europe. It is interesting to see that power plant emissions estimated by global/regional inventories are on averaged biased high by a factor of 6 as compared with those from unit-based databases. This supports our hypothesis

that emissions from some of these sources are distributed over too few point sources in global/regional inventories, as emissions from unit-based databases are expected to be more accurate due to the use of continuous emissions monitoring systems and unit-level fuel consumptions/emission factors (Liu et al., 2016).

## 6 Conclusions and Future Work

In this work we developed a merged emission inventory, OMI-HTAP, by combining OMI satellite-based emission estimates

for about 500 larger point sources (Fioletov et al., 2016) and a state-of-art bottom-up inventory HTAP v2.2 for smaller sources. Consistent with the HTAP inventory, the OMI-HTAP inventory provides monthly gridded $SO_2$ emissions with global coverage at a spatial resolution of $0.1\,°\times0.1\,°$. OMI-HTAP is available for the period from 2005 to 2014, but is most accurate for years 2008 and 2010, the years for which HTAP v2.2 was developed. We plan to include more recent years in the near future and use other bottom-up inventories in which multi-year estimates are provided.

The accuracy of OMI-HTAP has been evaluated by comparing modelled surface $SO_2$ concentrations with the measurements from ground-based air-quality monitoring networks focusing on the year 2010. GEOS-5 simulations using OMI-HTAP showed considerably better agreement with in situ measurements compared with those using the bottom-up inventory. The reduction in model bias is highlighted for the US, with the normalized mean bias decreasing from 0.41 (HTAP) to -0.03 (OMI-HTAP) for 2010. The improvements obtained with OMI for tracking emission changes over the years

2006-2014 is similarly confirmed by evaluation with ground-based data.

OMI-HTAP developed in this work has several advantages as compared with conventional bottom-up inventories. To our knowledge, it is the first inventory with inclusion of nearly 40 OMI-detected sources that are not included in previous





leading bottom-up inventories. It enables more accurate emission estimates for regions with such missing sources, e.g., the Middle East and Mexico. OMI-HTAP $SO_2$ emissions estimates for the Persian Gulf, Mexico, and Russia are 59%, 65%, and 56% higher than HTAP estimates in 2010, respectively. Unlike satellite observations, bottom-up inventories typically cannot provide high-quality local information on point sources for all countries. For instance, the European Union (EU) has reported

total $SO_2$ emissions for each country since decades, but the directive for reporting emissions from point sources with corresponding public database started in 2007 and the quality of data varies over EU countries. In developing countries, such data infrastructure has not been built up yet.

OMI-HTAP provides dynamic emissions for over 400 OMI-based large sources since 2005, allowing for updates to the emissions over time. Such update based on satellite measurements is more consistent than that compiled in bottom-up

inventories with annual activity statistics. The US, Europe, and China show declines of 47%, 27%, and 23% in the OMI-HTAP during 2005–2014, respectively.

Exact location of each large point source in OMI-HTAP is obtained from satellite observations and crosschecked by Google Earth manually. The location information contributes to correction of mislocated emissions arising from the downscaling approach adopted by bottom-up inventories or inaccurate locations provided by point source databases which

sometimes use the administrative or even postal address but not the coordinate of the stack as the location of the facility.

Although satellite data provide good information on the locations and trends for larger sources, they are currently not sufficient for providing complete information on $SO_2$ emissions and therefore much be merged with bottom-up inventories. We plan to combine satellite-based emission estimates with other bottom-up inventories in which multi-year estimates are provided, e.g., EDGAR v4.3.1 (Crippa et al., 2016) of the Joint Research Centre and the Community Emissions

Data System (CEDS; Hoesly et al., 2018) of Pacific Northwest National Laboratory, to better present emissions for small sources that cannot be detected by satellites or to use the historic trends for extrapolating backwards in time.

We anticipate that our approach can be used with higher spatial and temporal resolution satellite observations that will be available in the near future. This will complement and improve merged inventories by providing more accurate satellite-based emissions estimates, potentially with diurnal and seasonal variability. Improved global satellite observations

are anticipated from new sensors in low Earth orbit (LEO). The recently launched TROPOspheric Monitoring Instrument (TROPOMI) on the LEO ESA Sentinel-5 Precursor satellite (Veefkind et al., 2012) featuring approximately $7{\times}7$ $km^2$. The recently launched LEO NASA/NOAA JPSS-1/NOAA-20 OMPS instrument also has greater resolution (up to $10{\times}10$ $km^2$) than its predecessor on the NASA/NOAA Suomi National Polar-orbiting Partnership (SNPP) spacecraft ($50{\times}50$ $km^2$). Zhang et al. (2017) showed that higher spatial resolution observations increase the detection limit of $SO_2$ sources. This is

particularly important in the future, as emissions may continue to decrease due to emission control measures.

Finally, upcoming geostationary Earth orbiting (GEO) satellite instruments will enable emissions estimates for different times of the day at relatively high spatial resolution. Planned GEO atmospheric composition instruments include the Korean Geostationary Environmental Monitoring Spectrometer (GEMS; Kim et al., 2012), NASA Tropospheric



Emissions: Monitoring of Pollution (TEMPO; Chance et al., 2012), and ESA Sentinel-4 (Ingmann et al., 2012). These will have high spatial resolution similar to TROPOMI but on an hourly basis.

## Acknowledgements

This work was funded by NASA through the Aura and GMAO core programs. We thank the NASA Earth Science Division (ESD) Aura Science Team program for funding of OMI $SO_2$ product development and analysis (Grant # 80NSSC17K0240). We acknowledge the free use of the HTAP v2.2 and the EDGAR v4.3 emission inventories from European Commission, Joint Research Centre (JRC)/Netherlands Environmental Assessment Agency (PBL). We acknowledge TNO for providing the MACC-II emission inventory, Tsinghua University for providing the MIX emission inventory, and National Institute for Environmental Studies of Japan for providing the REAS 2 emission inventory. We thank the AQS, AirBase, and EANET networks for making their data available on line. We thank the KNMI and OMI SIPS team for providing and processing the OMI data and the GMAO's MERRA-2 team for the data sets used to drive the model simulations. We thank Dr. Mian Chin for helpful comments and Dr. Zifeng Lu for information on power plants in India.

## Data availability

The OMI-HTAP inventory is publicly available for the years 2005-2014 through the Aura Validation Data Center (AVDC) at https://avdc.gsfc.nasa.gov/pub/data/project/OMI_HTAP_emis/. The GEOS-5 model outputs are available upon request from the corresponding author.

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




## Figure

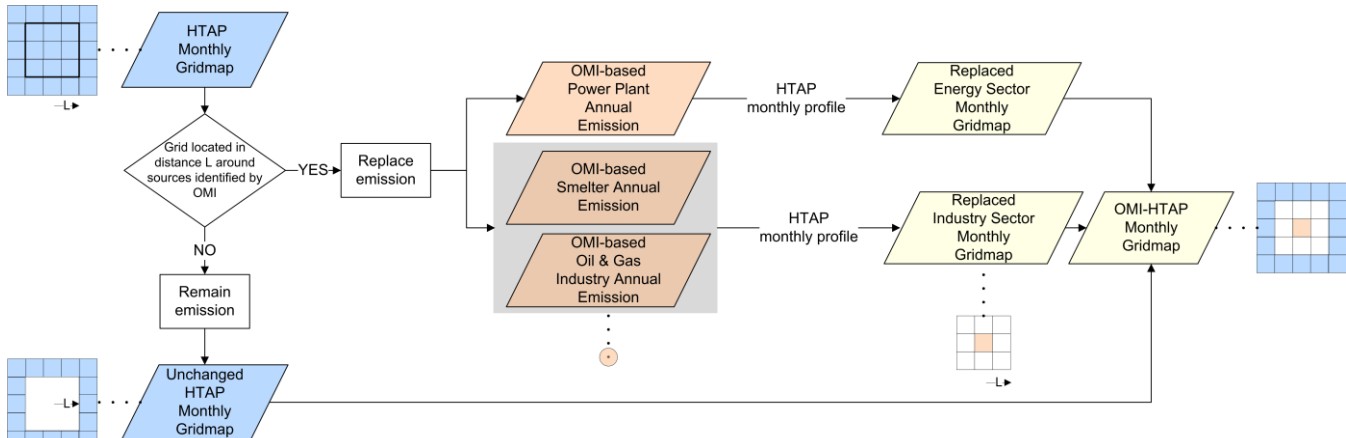

**Figure 1: Schematic methodology of the OMI-HTAP emission inventory development.**



**Figure 2: Maps for SO$_2$ emissions in the OMI-HTAP inventory, 2010 (a) and the differences between the OMI-HTAP and the HTAP inventory, 2010 (b). SO$_2$ emissions in the HTAP inventory are subtracted from those in the OMI-HTAP inventory to derive the differences. Emissions are regridded at the resolution of 1°×1° for illustration. The unit is Gg-SO$_2$ per grid cell.**





**Figure 3: Annual mean surface SO₂ concentration in 2010 based on the GEOS-5 model driven by the OMI-HTAP inventory, 2010 (a) and the differences between the modelled SO₂ using the OMI-HTAP and the HTAP inventory, 2010 (b). SO₂ concentrations using the HTAP inventory are subtracted from those in the OMI-HTAP inventory to derive the differences.**




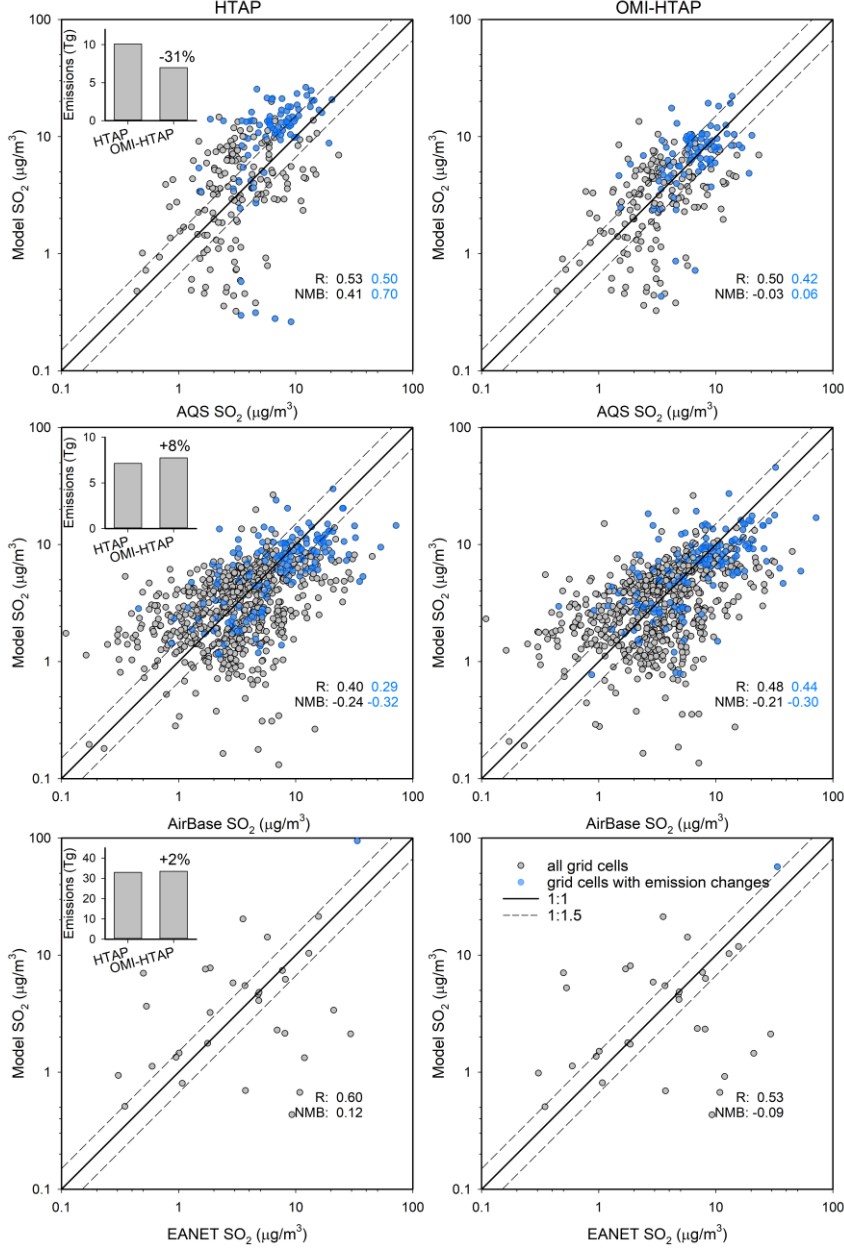

**Figure 4: Comparison of 2010 modelled and observed surface SO$_2$ concentrations. Observations are from the US AQS sites (top), European AirBase sites (middle), and East and Southeast Asia EANET sites (bottom). The annual averaged SO$_2$ concentrations are calculated for simulations using the HTAP (left) and the OMI-HTAP inventory (right). The blue dots denote the grid cells with differences in emissions between the HTAP and the OMI-HTAP inventories. The inset plots compare the total SO$_2$ emissions in those two inventories for the associated regions. The number on the top of the bars indicates the percentage of emission changes when comparing OMI-HTAP to HTAP. The values of correlation coefficient (R) and normalized mean bias (NMB) are colour coded by black and blue for all dots and blue dots, respectively.**





**Figure 5: Annual averaged SO₂ surface concentrations from AQS measurements in 2010 (a) and their differences between the modelled SO₂ using the HTAP (b) and the OMI-HTAP inventory, 2010 (c). AQS measurements are subtracted from the modelled SO₂ to derive the differences. The outline of circles corresponding to the grid cells with differences in emissions between the HTAP and the OMI-HTAP inventories is highlighted in black.**





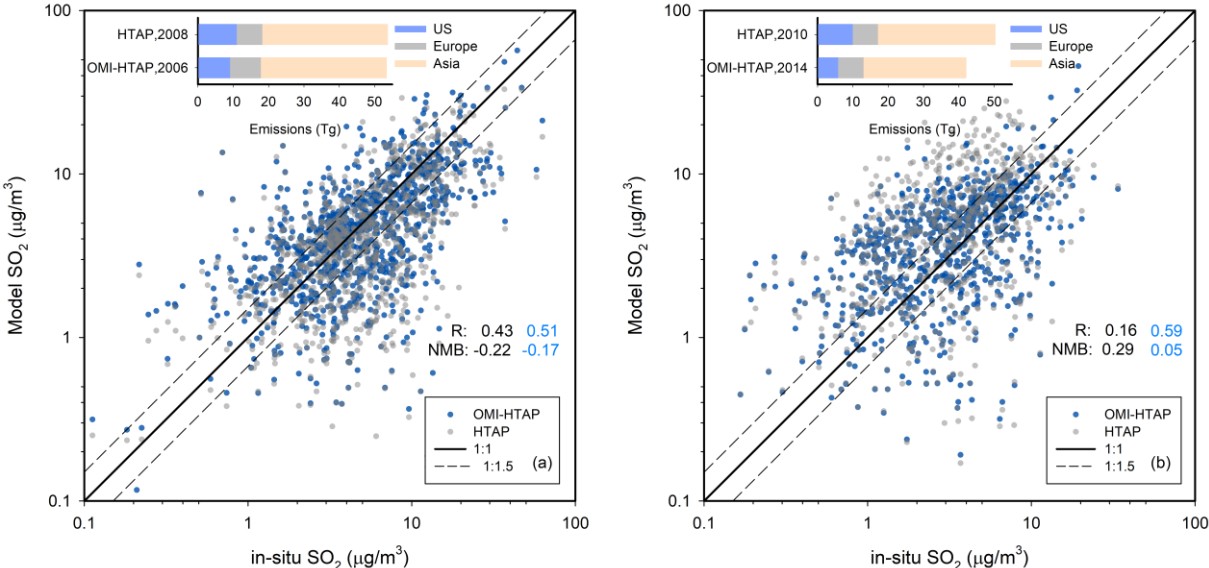

**Figure 6: Comparison of modelled annual averaged SO$_2$ surface concentrations for in-situ sites in 2006 (a) and 2014 (b). The grey and blue dots denote values using the HTAP and the OMI-HTAP inventories for corresponding years, respectively. The inset plots compare emissions from two inventories by region. Note that Europe only includes European countries with AirBase sites and Asia only includes East and Southeast Asia in the plot. The values of correlation coefficient (R) and normalized mean bias (NMB) are color coded by black and blue for grey and blue dots, respectively. Note that the plots use logarithmic scales, but R and NMB are calculated based on original data.**





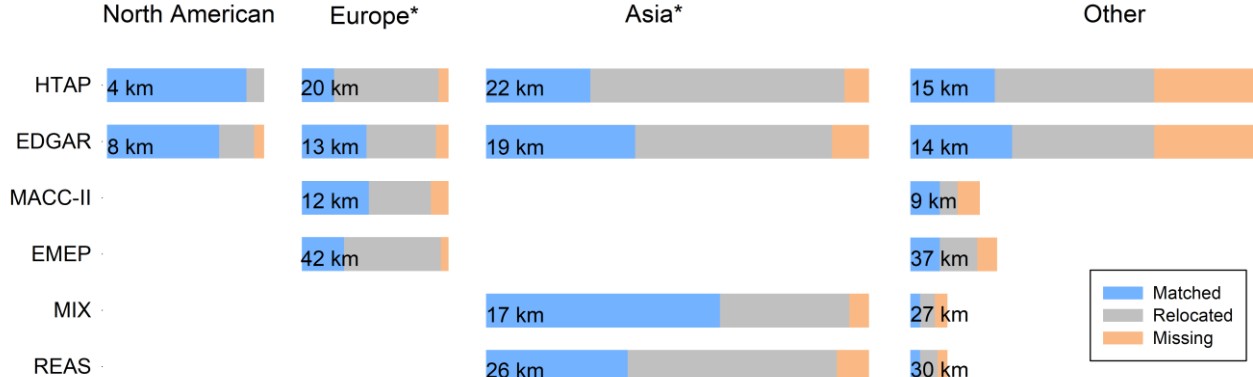

**Figure 7: Comparison of locations of anthropogenic large point sources detected by OMI with those in bottom-up inventories. The length of the bar denotes the number of sources. The blue bar denotes the sources with the same location in both the bottom-up and the OMI-based inventory (matched). The grey bar denotes the sources with location mismatches between the bottom-up and OMI-based inventories (relocated). The red bar denotes the OMI-based sources missing from the bottom-up inventory (missing). The numbers denote the average distance between OMI-detected locations and those in the bottom-up inventory for both relocated and matched sources.**

*Sources from countries that are only partly covered by European or Asian inventory, like Russia, Turkmenistan, Uzbekistan and Kazakhstan, are categorized as other to remain consistent with HTAP.









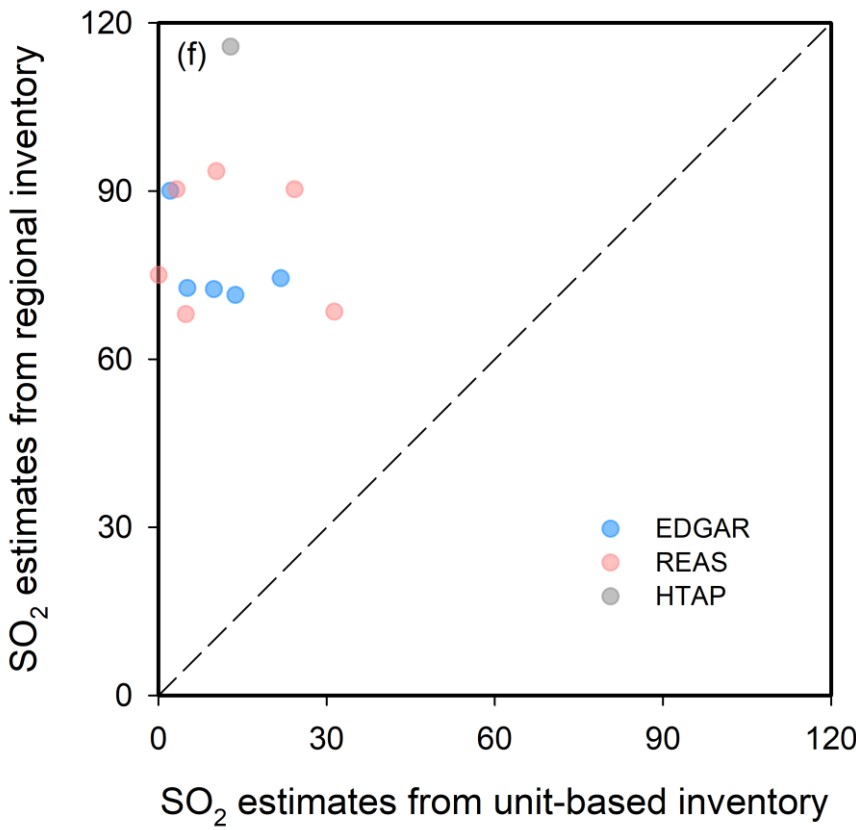

**Figure 8: (a) Geographic distribution of SO$_2$ sources in the OMI-based emission catalogue (Fioletov et al., 2011). SO$_2$ sources identified that were found to be missing from bottom-up inventories are in blue. Locations of large sources indicated by bottom-up inventories but not detected by OMI (unmatched) over (b) North America, (c) South America, (d) Europe, and (e) Asia. The background is the global mean SO$_2$ distribution (in DU) map for 2005–2014. The area affected by the South Atlantic Anomaly is shown as a white oval. (f) Comparison of SO$_2$ emission estimates from unit-based and regional emission inventory for power plants that are not detected by OMI.**

