# Peer review of "A new global anthropogenic $SO_2$ emission inventory for the last decade: A mosaic of satellite-derived and bottom-up emissions"

_Atmospheric Chemistry and Physics, 2018_

## Referee Comment (RC2)

Review ACP-2018-331

A new global anthropogenic SO$_2$ emission inventory for the last decade: A mosaic of satellite-derived and bottom-up emissions

By Liu et al.

This paper presents a much appreciated (much needed?) upgrade of global estimates of emissions of sulfur dioxide (SO2) using a combination of bottom-up information (HTAP) and satellite measurements (OMI).

Each source of information has its pros and cons, but not necessarily the same, so it is worthwhile to try to integrate them into a combined bottom-up/top-down SO2 emission inventory (HTAP-OMI).

The paper convincingly shows that, using the emission inventories in model simulations and comparing model results with independent observations of atmospheric SO2, that the combined OMI-HTAP emission database provides better results.

The study reveals many discrepancies related to slight mis-allocation of sources, as well as well-known issues with regard to representativeness of measurements, satellite data quality, and documented emission sources like underreported regions.

Overall, the approach is a step forward towards the integration of bottom-up information with top-down information. The method also provides a direction for increased and integrated use of upcoming new satellite missions to better estimate worldwide sources of emissions.

Overall, the paper is well written and well structured, and publishable after consideration of the remarks and suggestions below.

**[1] general remark**

I know the use of "higher/lower" is commonplace science publications when referring to other things than heights, but Personally, I prefer to use "larger/smaller" rather "higher/lower" when discussing anything unrelated to heights, in particular in atmospheric science, in order to avoid confusion.

I leave it up to the authors to decide, but if agreed, please check the entire paper and modify accordingly.

**[2] typos and suggestions for minor grammar changes.** Please note the remark regarding P9, L17-L19. Also note some suggestions are provided for improving a few figures.

P1, L31. Change to "We focus for the validation …"
P1, L32. Change to "… and for which a relatively large number of …"
P1, L33. Change to "… improves the agreement between the model and observations."
P1, L34-35. Suggest to move "Additionally, our … detected by satellites" to earlier in the abstract in L28, after "… with such missing sources.". Then, suggest to change "Additionally" to "In addition".
P1, L39/P2, L1-2. Suggest to move last line of abstract to after the previous suggestion (P1, L34-35).
P1, L37. Remove "For example,"

P2, L13. "earlier" could be made more specific (SO2 emission regulations in the US and EU are introduced in the 1980s)
P2, L21. I think it should be "removal", not "removals"
P2, L21. Suggest to change to "The spatial distribution of emissions is even more uncertain"
P2, L21-22. Change to "… are in most cases allocated by spatial proxies rather than …"
P2, L23-L24. Move "developing" to the beginning of the sentence: "In addition, developing …"
P2, L26. Replace "indicate" with "identify"

P3, L5. "Gaussian distributions" or "Gaussian dispersion"
P3, L11-L12. I think what is meant here is "that combines information about large SO2 sources from …"

P4, L3-5. Please provide a brief justification or explanation (or reference) for the source strength dependence on parameter L.
P4, L7. Unclear where "In this way …" refers to.

P5, L1. "monitoring systems" (plural)
P5, L20. Latest: later, more recent?
P5, L24. Replace "it is located" with for example "emissions are located" or something similar
P5, L30. Explain what is meant here with "profile". In atmospheric sciences, a "profile" generally refers to a vertical distribution of a parameter.
P5, L33. "OMI-HTAP inventory"

P6, L4-L5. Delete "On the contrary,"
P6, L24. Change "less" to "smaller"
P6, L33. Maybe replace "reported" with "documented" to avoid double use of "report" in the sentence?

P7, L1. "Long standing experience", would still be nice to add some references.

P9, L14. "identifies" maybe change to "shows" or "reveals"?
P9, L17-19. I think it would be nice to mention here (or in the conclusions) that OMI-HTAP changes 2008-2008 are more consistent with EPA estimates than HTAP. Important finding.

P11, L3. Change "knowledge of" to "information about"

P12, L1. "my" should be "may"
P12, L6. Replace "less" with "fewer"
P12, L14. "averaged" to "average"
P12, L31. Maybe use "The OMI-HTAP emission database developed …"?

P13, L1. "leading", maybe "widely used"?
P13, L5. "since decades" to "for many decades"?
P13, L12. "The exact location of …"

**[3] figures**

Figure 2, upper panel. Most land areas are color coded in dark blue/purple. Does this include the 0.0 value, or does this mean that emissions indeed are larger than zero. If not zero, this also begs the question how SO2 emission over most land areas are larger than zero. If indeed zero, it is worth considering color coding land areas with zero SO2 emissions in grey or white. Please clarify in caption and/or modify the figure.

Figure 2, lower panel. Is not necessary insightfull because of the small pixels. Is it possible to use symbols like a filled circle that for example indicate the size of the difference? I don't know if that would work, but I think it is worth trying to see if that helps for the usefulness of the panel. Also, it might be worth to color code non-zero differences between +/- 5 Gg/yr/grid with greys (see discussion about the upper panel). As the panel now stands, there is no separation between zero emission locations and locations with small differences.

Figure 3, upper panel. Add whites for (near) zero concentrations, as now everything is mostly blue, not very appealing. And similar to previous discussion, maybe separate small concentrations from zero concentrations.

---

## Referee Comment (RC1) · Anonymous Referee #1 · 31 Jul 2018

This manuscript developed a new global SO2 emission inventory by integration of bottom-up inventory and satellite observations. Satellite-based observations have been widely used in providing top-down constraints on surface emissions; however, top-down inventories are difficult to be used in due to lack of bottom-up information such as sectoral contribution. This work developed a harmonization approach that integrated OMI-inferred emission information into HTAP global emission inventory, and the new inventory has been proved to improve the model agreement with observations. The new method developed from this work has large potential in improving and timely updating bottom-up inventories. This is a very timely work for the emission inventory community. It's novel, and relevant to ACP readership. This manuscript is clearly structured and generally well written. It could be published in ACP after addressing the following minor issues.

1. A direct comparison (scatter plots) between OMI-based estimates and bottom-up inventory should be provided over the locations where OMI estimates are available;

2. Uncertainties of OMI-based estimates should be discussed comprehensively and compared with bottom-up inventories;

3. It would be nice if the authors could provide some insights of using this approach for other pollutants such as NOx.

---

## Author Comment (AC1) · 17 Sep 2018

*This manuscript developed a new global SO$_2$ emission inventory by integration of bottom-up inventory and satellite observations. Satellite-based observations have been widely used in providing top-down constraints on surface emissions; however, top-down inventories are difficult to be used in due to lack of bottom-up information such as sectoral contribution. This work developed a harmonization approach that integrated OMI-inferred emission information into HTAP global emission inventory, and the new inventory has been proved to improve the model agreement with observations. The new method developed from this work has large potential in improving and timely updating bottom-up inventories. This is a very timely work for the emission inventory community. It's novel, and relevant to ACP readership. This manuscript is clearly structured and generally well written. It could be published in ACP after addressing the following minor issues.*

**Response:** We thank Referee #1 for the encouraging comments. All comments and suggestions have been considered carefully and well addressed below.

*1. A direct comparison (scatter plots) between OMI-based estimates and bottom-up inventory should be provided over the locations where OMI estimates are available.*

**Response:** We thank you for the suggestions. We have added the illustration for differences between OMI-based estimates and bottom-up inventory in Figure 2b. We have also added a scatter plot (Figure S1) comparing the satellite-derived and the HTAP emissions estimates for year 2010 in the supplement.

*2. Uncertainties of OMI-based estimates should be discussed comprehensively and compared with bottom-up inventories.*

**Response:** We thank you for the suggestions. We have added the discussion about uncertainties of OMI-based emissions estimates and compared it with bottom-up inventories in Section 2.3, as follows:

"Uncertainties in the OMI-based estimates may contribute to the differences. These uncertainties can be grouped into three categories: in the retrieval of the OMI SO$_2$ vertical column density (VCD); those that come from the fit of the OMI-detected SO$_2$ downwind plume; and those related to the wind information. The overall uncertainty in annual emissions is estimated to be around 50% (Fioletov et al., 2016), with the primary contributors of the air mass factor

calculation when determining VCD (27%) and the wind height (20%). On the other hand, uncertainties inherent in the total magnitude of bottom-up emissions may also contribute to the differences such as when bottom-up emissions are not routinely updated. The uncertainties of emissions from the industry sector are estimated to range from 15% to 70% over countries depending on how well the statistical infrastructure is maintained by individual countries (Janssens-Maenhout et al., 2015 and references in there). In addition, the uncertainties of spatial distribution may cause the differences. In fact, emissions from some emitting sectors in bottom-up inventories are not tracked with individual point sources but spread out over larger areas instead. The country-specific emissions in HTAP are allocated where possible to the locations of point sources (e.g. public electricity plants), but a large fraction (e.g. some smelters of which the location are not available) remains distributed over the countries with spatial proxies (e.g., urban population) of which the representativeness is only qualitatively known."

*3. It would be nice if the authors could provide some insights of using this approach for other pollutants such as $NO_x$.*

**Response:** We thank you for the suggestions. We have added the discussion about the insights of the approach in conclusion, as follows:

"Finally, the merging inventory methodology proposed in this study is potentially applicable for other air pollutants. It has good potential for application to $NO_x$, as $NO_x$ emissions from power plants and cities can be quantified by similar CTM-independent approaches as well [Beirle et al., 2011; Liu et al., 2016]. However, merging satellite-derived urban $NO_x$ estimates with bottom-up inventories is more challenging than point source emissions. Urban emissions are distributed over a larger number of sectors, including large contributions from areal sources such as road transport. An alternative method needs to be explored to reconcile bottom-up and top-down satellite-derived urban emissions."

---

## Author Comment (AC2) · 17 Sep 2018

*This paper presents a much appreciated (much needed?) upgrade of global estimates of emissions of sulfur dioxide (SO2) using a combination of bottom-up information (HTAP) and satellite measurements (OMI).*

*Each source of information has its pros and cons, but not necessarily the same, so it is worthwhile to try to integrate them into a combined bottom-up/top-down SO2 emission inventory (HTAP-OMI).*

*The paper convincingly shows that, using the emission inventories in model simulations and comparing model results with independent observations of atmospheric SO2, that the combined OMI-HTAP emission database provides better results.*

*The study reveals many discrepancies related to slight mis-allocation of sources, as well as well-known issues with regard to representativeness of measurements, satellite data quality, and documented emission sources like underreported regions.*

*Overall, the approach is a step forward towards the integration of bottom-up information with top-down information. The method also provides a direction for increased and integrated use of upcoming new satellite missions to better estimate worldwide sources of emissions.*

*Overall, the paper is well written and well structured, and publishable after consideration of the remarks and suggestions below.*

**Response:** We thank Referee #3 for the encouraging comments. All comments and suggestions have been considered carefully and well addressed below.

*[1] general remark*

*I know the use of "higher/lower" is commonplace science publications when referring to other things than heights, but Personally, I prefer to use "larger/smaller" rather "higher/lower" when discussing anything unrelated to heights, in particular in atmospheric science, in order to avoid confusion.*

*I leave it up to the authors to decide, but if agreed, please check the entire paper and modify accordingly.*

**Response:** We thank you for the suggestions. We have checked the entire paper and modified it accordingly.

*[2] typos and suggestions for minor grammar changes. Please note the remark regarding P9, L17-L19. Also note some suggestions are provided for improving a few figures.*

*P1, L31. Change to "We focus for the validation …"*

*P1, L32. Change to "… and for which a relatively large number of …"*

*P1, L33. Change to "… improves the agreement between the model and observations."*

*P1, L34-35. Suggest to move "Additionally, our … detected by satellites" to earlier in the abstract in L28, after "… with such missing sources.". Then, suggest to change "Additionally" to "In addition".*

*P1, L39/P2, L1-2. Suggest to move last line of abstract to after the previous suggestion (P1, L34-35).*

*P1, L37. Remove "For example,"*

**Response:** Thanks. We have corrected them based on above comments in the revised manuscript.

*P2, L13. "earlier" could be made more specific ($SO_2$ emission regulations in the US and EU are introduced in the 1980s)*

**Response:** Thanks. We have changed it in the revised manuscript, as follows:

"in particular earlier (since the 1980s for power plants) in the US and Europe (Crippa et al., 2016)".

*P2, L21. I think it should be "removal", not "removals"*

*P2, L21. Suggest to change to "The spatial distribution of emissions is even more uncertain"*

*P2, L21-22. Change to "… are in most cases allocated by spatial proxies rather than …"*

*P2, L23-L24. Move "developing" to the beginning of the sentence: "In addition, developing …"*

*P2, L26. Replace "indicate" with "identify"*

*P3, L5. "Gaussian distributions" or "Gaussian dispersion"*

*P3, L11-L12. I think what is meant here is "that combines information about large SO2 sources from …"*

**Response:** Thanks. We have corrected them based on above comments in the revised manuscript.

*P4, L3-5. Please provide a brief justification or explanation (or reference) for the source strength dependence on parameter L.*

**Response:** Thanks. We add the reference of Fioletov et al.(2016) in the revised manuscript.

*P4, L7. Unclear where "In this way …" refers to.*

**Response:** We agree that the phrase is confusing. We deleted the sentence in the revised manuscript.

*P5, L1. "monitoring systems" (plural)*

*P5, L20. Latest: later, more recent?*

*P5, L24. Replace "it is located" with for example "emissions are located" or something similar*

**Response:** Thanks. We have corrected them based on above comments in the revised manuscript.

*P5, L30. Explain what is meant here with "profile". In atmospheric sciences, a "profile" generally refers to a vertical distribution of a parameter.*

**Response:** We changed "monthly profile" to "monthly variations" in the revised manuscript.

*P5, L33. "OMI-HTAP inventory"*

*P6, L4-L5. Delete "On the contrary,"*

*P6, L24. Change "less" to "smaller"*

*P6, L33. Maybe replace "reported" with "documented" to avoid double use of "report" in the sentence?*

**Response:** Thanks. We have corrected them based on above comments in the revised manuscript.

*P7, L1. "Long standing experience", would still be nice to add some references.*

**Response:** We add the reference of Hoesly et al. (2018a) in the revised manuscript.

*P9, L14. "identifies" maybe change to "shows" or "reveals"?*

**Response:** We changed it to "reveals".

*P9, L17-19. I think it would be nice to mention here (or in the conclusions) that OMI-HTAP changes 2008-2008 are more consistent with EPA estimates than HTAP. Important finding.*

**Response:** We pointed out the better consistency in the revised manuscript, as follows:

"The magnitude of $SO_2$ emissions decreases by 25% in OMI-HTAP during 2008 to 2010, consistent with the decline of 25% reported by EPA (EPA Air Pollutant Emissions Trends Data; available at https://www.epa.gov/air-emissions-inventories/air-pollutant-emissions-trends-data) and much larger than the decline of 9% in HTAP."

*P11, L3. Change "knowledge of" to "information about"*

*P12, L1. "my" should be "may"*

*P12, L6. Replace "less" with "fewer"*

*P12, L14. "averaged" to "average"*

*P12, L31. Maybe use "The OMI-HTAP emission database developed ..."?*

*P13, L1. "leading", maybe "widely used"?*

*P13, L5. "since decades" to "for many decades"?*

*P13, L12. "The exact location of ..."*

**Response:** Thanks. We have corrected them based on above comments in the revised manuscript.

*[3] figures*

*Figure 2, upper panel. Most land areas are color coded in dark blue/purple. Does this include the 0.0 value, or does this mean that emissions indeed are larger than zero. If not zero, this also begs the question how $SO_2$ emission over most land areas are larger than zero. If indeed zero, it is worth considering color coding land areas with zero $SO_2$ emissions in grey or white. Please clarify in caption and/or modify the figure.*

**Response:** The legend does not include the 0.0 value. We have clarified this in the revised caption.

The figures below display the maps for HTAP $SO_2$ emissions from the energy (top) and the residential (bottom) sector for year 2010 at the resolution of $0.1° \times 0.1°$ (available at http://edgar.jrc.ec.europa.eu/htap_v2/index.php). In principle, emissions are spatially distributed based on their locations. As shown in the map for the energy sector (top), emissions from large point sources like power plants are allocated to grid cells based on their coordinates and thus there are many land areas showing no emissions. However, this is not the case for sectors with areal sources as dominated sources, like the residential sector (bottom). Emissions are distributed to grid cells using spatial proxy due to the lack of information on locations and thus most land

areas are distributed to emissions. In addition, emissions are regridded at much lower resolution $(1° \times 1°)$ for illustration in Figure 2. It further improves the possibility of being distributed to emissions for individual grid cells, as the size of grid cells is larger.

[Figure]

*Figure 2, lower panel. Is not necessary insightfull because of the small pixels. Is it possible to use symbols like a filled circle that for example indicate the size of the difference? I don't know if that would work, but I think it is worth trying to see if that helps for the usefulness of the panel. Also, it might be worth to color code non-zero differences between +/- 5 Gg/yr/grid with greys (see discussion about the upper panel). As the panel now stands, there is no separation between zero emission locations and locations with small differences.*

**Response:** Thanks. We changed the symbol accordingly and color coded small differences with light yellow in the revised figure. The areas without emissions changes are separated by grey as well.

*Figure 3, upper panel. Add whites for (near) zero concentrations, as now everything is mostly blue, not very appealing. And similar to previous discussion, maybe separate small concentrations from zero concentrations.*

**Response:** Thanks. There are no grid cells with zero concentrations at the modelling resolution of $0.5°\times0.5°$. We have added whites for near zero concentrations in the revised figure.